# N^6^-Methyladenosine Directly Regulates CD40L Expression in CD4^+^ T Lymphocytes

**DOI:** 10.3390/biology12071004

**Published:** 2023-07-14

**Authors:** Ellen C. N. van Vroonhoven, Lucas W. Picavet, Rianne C. Scholman, Noortje A. M. van den Dungen, Michal Mokry, Anouk Evers, Robert J. Lebbink, Jorg J. A. Calis, Sebastiaan J. Vastert, Jorg van Loosdregt

**Affiliations:** 1Center for Translational Immunology, University Medical Center Utrecht, 3584 CX Utrecht, The Netherlands; e.c.n.vanvroonhoven@umcutrecht.nl (E.C.N.v.V.); l.w.picavet@umcutrecht.nl (L.W.P.); r.c.scholman@umcutrecht.nl (R.C.S.); j.j.a.calis-2@umcutrecht.nl (J.J.A.C.); b.vastert@umcutrecht.nl (S.J.V.); 2Department of Experimental Cardiology, University Medical Center Utrecht, 3584 CX Utrecht, The Netherlands; n.a.m.vandendungen@umcutrecht.nl (N.A.M.v.d.D.); m.mokry@umcutrecht.nl (M.M.); 3Department of Medical Microbiology, University Medical Center Utrecht, 3584 CX Utrecht, The Netherlands; a.evers-5@umcutrecht.nl (A.E.); r.j.lebbink-2@umcutrecht.nl (R.J.L.); 4Department of Pediatric Rheumatology and Immunology, University Medical Center Utrecht, 3584 CX Utrecht, The Netherlands

**Keywords:** RNA methylation, epitranscriptomics, CD40 ligand, T cell activation, adaptive immunity, autoimmunity

## Abstract

**Simple Summary:**

The tight regulation of the expression of cytokines, surface receptors and co-stimulatory and co-inhibitory molecules is necessary for a well-functioning immune system. There are various cellular processes that regulate the expression of these immune regulatory proteins, for instance at the RNA transcript level. Epitranscriptomic regulation, involving RNA binding proteins and modifications, can determine the turnover and translation of mRNA transcripts. N^6^-methyladenosine (m^6^A) is the most abundant RNA modification in eukaryotic cells and it was demonstrated that m^6^A is involved in the early differentiation of CD4^+^ T lymphocytes. However, the function of m^6^A in CD4^+^ T cell activation and function is still incompletely understood. Here, we demonstrate that m^6^A regulates the activation of CD4^+^ T lymphocytes via the regulation of CD40 ligand expression, a key co-stimulatory molecule expressed on the cell surface of CD4^+^ T cells. The discovery of this novel function of m^6^A and its regulatory proteins contributes to our general understanding of CD4^+^ T cell activation, gene expression regulation and autoimmune disease pathogenesis.

**Abstract:**

T cell activation is a highly regulated process, modulated via the expression of various immune regulatory proteins including cytokines, surface receptors and co-stimulatory proteins. N^6^-methyladenosine (m^6^A) is an RNA modification that can directly regulate RNA expression levels and it is associated with various biological processes. However, the function of m^6^A in T cell activation remains incompletely understood. We identify m^6^A as a novel regulator of the expression of the CD40 ligand (CD40L) in human CD4^+^ lymphocytes. Manipulation of the m^6^A ‘eraser’ fat mass and obesity-associated protein (FTO) and m^6^A ‘writer’ protein methyltransferase-like 3 (METTL3) directly affects the expression of CD40L. The m^6^A ‘reader’ protein YT521-B homology domain family-2 (YTHDF2) is hypothesized to be able to recognize and bind m^6^A specific sequences on the *CD40L* mRNA and promotes its degradation. This study demonstrates that CD40L expression in human primary CD4^+^ T lymphocytes is regulated via m^6^A modifications, elucidating a new regulatory mechanism in CD4^+^ T cell activation that could possibly be leveraged in the future to modulate T cell responses in patients with immune-related diseases.

## 1. Introduction

CD4^+^ T cell activation is achieved upon the binding of the peptide major histocompatibility complex (pMHC) on the surface of antigen-presenting cells (APCs) to the T cell receptor (TCR). Upon the pMHC–TCR interaction, co-stimulatory molecules such as CD28, inducible T cell co-stimulator (ICOS) and CD40 ligand (CD40L) are translocated to the CD4^+^ T cell surface. Together, these co-stimulatory and co-inhibitory molecules modulate the duration and intensity of TCR-mediated signaling, thereby facilitating an optimal immune response. Given the complexity of the immune response, precise regulation of the expression of co-stimulatory molecules amongst other processes is essential for a well-orchestrated T cell response.

Post-transcriptional regulation, involving RNA binding proteins and RNA modifications, plays a crucial role in the regulation of protein expression. One of these modifications is N^6^-methyladenosine (m^6^A), which was discovered in the 1970s and is the most prevalent RNA modification in eukaryotes [1,2,3]. m^6^A is a dynamic modification that is co- and post-transcriptionally installed on specific mRNA sequences, mostly located in the 3′-untranslated region (UTR), near the STOP codon and in the 5′-UTR of the processed mRNA [3]. m^6^A is increased in so-called ‘DRACH’-motif sequences (D = A, G, or U; R = G or A; H = A, C or U) [4] and is installed by a complex of ‘writer’ proteins that include METTL3 [5,6], methyltransferase-like 14 (METTL14) and Wilms’ tumor 1-associating protein (WTAP) [6]. ‘Reader’ proteins from the YT521-B homology domain-containing (YTH) family selectively bind to the m^6^A modification, thereby determining the fate of the transcript [7,8,9]. These readers have been described to regulate RNA stability and therefore RNA degradation (YTHDF1, YTHDF2 and YTHDF3) [9,10] by promoting transport towards processing bodies, or to be involved in splicing regulation and the nuclear export of mRNA transcripts (YTHDC1) [11]. m^6^A can be removed from the RNA by ‘eraser’ proteins FTO [12,13] or AlkB homologue 5 (ALKBH5) [14].

Recently, it has been demonstrated that m^6^A modifications are involved in early CD4^+^ T cell differentiation. For instance, METTL3 was identified as an important stabilizer of T follicular helper (T_FH_) cell signature gene *Tcf7*, indicating a role for m^6^A in T_FH_ cell differentiation [15]. Additionally, METTL3 was found to regulate naïve T lymphocyte proliferation and differentiation [16]. Naïve T cells from *Mettl3*-KO mice showed increased expression of *Socs1*, *Socs3* and *Cish*, thereby suppressing IL-7/STAT5 signaling and enhancing ERK and AKT signaling. The eraser protein ALKBH5 was demonstrated to have a checkpoint role in the early differentiation of γδ T cells and αβ T cells by targeting Jagged1/Notch2 signaling [17]. Despite recent advances in understanding the implications of m^6^A in CD4+ T cell regulation, the full implications of its involvement remain incompletely understood.

To obtain further insight into the role of m^6^A in T cell activation, we assessed the changes in the m^6^A modification of TCR signaling and T cell activation-associated RNA transcripts in human primary CD4^+^ T lymphocytes and identified many changes upon activation. m^6^A enrichment was found to be increased on co-stimulatory molecule CD40 ligand (*CD40L*) mRNA, primarily in the 3′ UTR of the transcript. Promoting m^6^A levels in primary CD4^+^ T cells resulted in the decreased expression of CD40L. YTHDF2, a m^6^A reader associated with RNA degradation, is hypothesized to directly bind to m^6^A-specific sequences in the 5′ and 3′-UTR of the *CD40L* mRNA. Taken together, our data demonstrate that the expression of CD40L in CD4^+^ T lymphocytes is directly regulated via m^6^A and its regulatory proteins.

## 2. Materials and Methods

### 2.1. Cell Culture

Peripheral blood mononuclear cells (PBMCs) were obtained from healthy donor blood using Ficoll-Paque Plus (Cytiva, Marlborough, MA, USA) density gradient media. Primary human CD4^+^ T lymphocytes were isolated via magnetic-activated cell sorting (MACS) using the human CD4^+^ T cell isolation kit (Miltenyi Biotec, Bergisch Gladbach, Germany). Cells were cultured in a Roswell Park Memorial Institute (RPMI) 1640 medium (Fisher Scientific, Hampton, NH, USA) containing 10% heat-inactivated human AB serum, 1% L-glutamine (Life Technologies, Carlsbad, CA, USA), 100 U/mL of penicillin and 100 mg/mL of streptomycin (Gibco, New York, NY, USA) supplemented with 50 U/mL of IL-2. Jurkat cells (Clone E6.1, ATCC, Manassas, VA, USA) were cultured in Dulbecco’s modified Eagle medium (DMEM) with GlutaMax (Gibco), supplemented with 10% heat-inactivated fetal calf serum (Sigma-Aldrich, St. Louis, MI, USA), 100 U/mL of penicillin and 100 mg/mL of streptomycin (Gibco). All cells were cultured at 37 °C in 5% CO_2_. For the manipulation of FTO and METTL3, 50 µM entacapone (Sigma, SML0654-10MG) and 30 µM STM2457 (MedChemExpress, Monmouth Junction, NJ, USA, 2499663-01-1) were used. For CRISPR/Cas9 genome engineering, single-guide RNAs (sgRNAs) were introduced in the lentiviral pSicoR-CRISPR-PuroR vector (RP-557) [18] as described previously. The used crRNA sequence for FTO knockout was GGCTGCTTATTTCGGGACC. CRISPR/Cas9 knockout of *FTO* in Jurkat cells was performed via lentiviral transduction. Second-generation lentiviral particles were produced in HEK293T cells using polyethylenimine (PEI) MAX (Polysciences Europe GmbH, Hirschberg, Germany), and Jurkat cells were transduced using 5 µg/mL of polybrene (Santa Cruz, Santa Cruz, CA, USA). Transduced cells were single-cell-cultured and selected with 2 µg/mL of puromycin (Merck, Rahway, NJ, USA).

### 2.2. meRIP Sequencing and qPCR

CD4^+^ T lymphocytes were isolated from healthy donor PBMCs using MACS. The cells were either activated for 4 h with PMA (20 ng/mL, Sigma) and ionomycin (1 ug/mL, Calbiochem), or stimulated overnight with CD3/CD28 Dynabeads (Invitrogen, Waltham, MA, USA). Cells were lysed in Trizol LS reagent (Fisher Scientific) and RNA was isolated according to the manufacturer’s protocol. For meRIPseq only, mRNA was isolated and subsequently fragmented by using the NEXTflex Rapid directional mRNA-seq bundle (5138-10). Subsequently, Protein A/G magnetic beads (Fisher Scientific) were coated with mouse anti-m^6^A (Synaptic Systems Gmbh, Göttingen, Germany, 202-111) or mouse anti-IgG2b,k isotype control (BD, 556577). mRNA was added to the coated beads and IP was performed. mRNA was washed using a high-salt–low-salt washing method. To elute the bound RNA, the beads were mixed with RLT lysis buffer from the RNeasy mini-kit (Qiagen, Hilden, Germany). This kit was subsequently used for the purification of the mRNA. Sequences from the Epimark N6-Methyladenosine Enrichment kit (New England Biolabs, Ipswich, MA, USA) were added to the samples. meRIPseq samples were single-end sequenced at 50 bp on the Illumina NextSeq500 sequencer (Utrecht DNA Sequencing Facility, Utrecht, The Netherlands). Sequencing reads were aligned using STAR (version 2.7.1) [19] to the human genome (version GRCh38) with transcriptomic information from ENSEMBL (version GRCh38.v100) supplemented with sequences from the Epimark N6-Methyladenosine Enrichment kit (New England Biolabs). Unique reads were selected, and expression was quantified at the gene level using htseq-count (version 0.11.4) [20]. Differential expression was analyzed with edgeR [21], and per sample expression was normalized using TMM normalization. cDNA synthesis was performed on the meRIP qPCR samples using the iScript cDNA synthesis kit (Bio-Rad, Hercules, CA, USA). qPCR was performed with SYBR Select mastermix (Life Technologies) in a Quantstudio 12K Flex (Thermo Fisher Scientific, Waltham, MA, USA) in accordance with the manufacturer’s protocol. A list of primers used in this study can be found in Appendix A.

### 2.3. Flow Cytometry

Cells were washed in FACS buffer (PBS with 2% FBS, and 0.1% NaN3), fixated and permeabilized with a fixation/permeabilization solution kit (BD) and stained with antibodies (Appendix A). Measurements were performed using the BD FACSCanto™ II flow cytometer, and FlowJo v10 was used for data analysis.

### 2.4. Western Blot

Cells were lysed in Laemmli buffer (0.12 M Tris-HCl, pH 6.8, 4% SDS, 20% glycerol, 0.05 µg/µL of bromophenol blue, and 35 mM β-mercaptoethanol). Samples were separated using SDS-PAGE on 10% gel and transferred to a polyvinylidene difluoride (PVDF) membrane (Merck). After blocking with 5% milk powder in 1% tris-buffered saline and Tween 20 (TBST), the membrane was probed with the antibodies indicated in Appendix A and analyzed using enhanced chemiluminescence (Thermo Fisher Scientific).

### 2.5. Statistics

Statistical analysis was performed using Graphpad Prism 9. The statistical tests used to test significance are specified in the figure legends.

## 3. Results

### 3.1. m^6^A Enrichment Is Increased on CD40L mRNA in Activated CD4^+^ T Lymphocytes

To assess the transcriptome-wide m^6^A landscape in activated human CD4^+^ lymphocytes, meRIP sequencing was performed on primary CD4^+^ T lymphocytes that were stimulated overnight with CD3/CD28 Dynabeads. m^6^A enrichment was increased on the mRNA transcripts of 687 genes, indicated by a log2 fold change value equal to or higher than 1.5. m^6^A methylated transcripts including many transcripts associated with T cell activation, in particular mRNA transcripts encoding for co-stimulatory proteins (Figure 1A). We specifically focused on CD40L for further assessment, as *CD40L* was highly expressed in the sequencing data and *CD40L* mRNA was highly enriched after the stimulation of T cells. Furthermore, CD40L was selected as it is an important co-stimulatory protein for TCR signaling and T cell activation, and there is evidence that CD40L is implicated in autoimmune disease pathogenesis [22,23,24]. In the meRIP-seq data, *CD40L* was highly expressed and methylated, and m^6^A methylation was increased mostly at the 5′- and 3′ UTR upon T cell activation (Figure 1B). To further evaluate *CD40L* and to confirm these results, meRIP-qPCR was performed. Again, m^6^A was found to be significantly enriched on the *CD40L* mRNA of stimulated CD4^+^ T cells compared to that on unstimulated cells (Figure 1C). To further validate these data, we assessed the m^6^A methylation of *CD40L* in the mouse CD4^+^ T cell m^6^A CLIP-seq dataset that was made available by Ito-Kureha et al. [25]. In line with our observations in human CD4^+^ T cells, *CD40L* was shown to be methylated at the 3′-UTR (Figure 1D). Taken together, these data demonstrate that *CD40L* mRNA can be m^6^A methylated and that the methylation of these transcripts increases upon CD4^+^ T cell activation.

### 3.2. CD40L Expression Is Regulated via m^6^A ‘Eraser’ FTO in Activated CD4^+^ T Lymphocytes

m^6^A is often associated with (increased) mRNA degradation [9]. To determine whether or not m^6^A regulates CD40L expression, m^6^A levels were manipulated via the CRISPR/Cas9-mediated knockout of the demethylase FTO in a Jurkat cell line (Figure 2A). After knockout, cells were stimulated and *CD40L* mRNA expression was assessed using qPCR. FTO knockout reduced *CD40L* mRNA expression levels indicating that the increased m^6^A methylation of *CD40L* promotes its degradation (Figure 2B). Subsequently, CD40L protein expression was assessed using flow cytometry upon stimulation (Figure 2C and Appendix A). Both the percentage of cells expressing CD40L and the amount of CD40L expressed per cell was decreased upon the knockout of FTO. To assess whether or not m^6^A also regulates CD40L expression in primary cells, healthy control CD4^+^ T lymphocytes were treated with an FTO inhibitor [26] and CD40L protein expression was assessed (Figure 2D and Appendix A). While TCR stimulation increased CD40L expression, both the amount of CD40L-expressing cells as well as the levels of CD40L expressed by each cell was decreased after incubation with the FTO inhibitor. Again, these observations were reflected by similar changes induced in mRNA levels, as the amount of *CD40L* mRNA was also decreased upon FTO inhibition (Figure 2E). The expression of *CD40L* pre-mRNA did not change upon FTO inhibition, indicating that the observed changes in CD40L expression are the result of the direct degradation of CD40L and not the result of decreased *CD40L* transcription (Figure 2F). Together, these results indicate that CD40L expression in activated CD4^+^ T lymphocytes is directly regulated by m^6^A and its eraser protein FTO.

### 3.3. METTL3 Regulates CD40L Expression in Activated CD4^+^ T Lymphocytes

To further validate that m^6^A is involved in the regulation of CD40L expression, human CD4^+^ T lymphocytes were treated with an inhibitor of METTL3 [27,28,29,30], thereby reducing m^6^A levels. Flow cytometry was performed to assess CD40L protein expression (Figure 3). As observed previously, CD40L expression was increased upon the stimulation of the CD4^+^ T lymphocytes. Incubation with the METTL3 inhibitor resulted in an increased expression of both the amount of CD40L-expressing cells and the expression of CD40L per cell. Taken together, these results indicate that CD40L expression in stimulated CD4^+^ T lymphocytes is regulated via m^6^A.

### 3.4. m^6^A Reader YTHDF2 Possiblybinds to m^6^A Sequences on Cd40lg mRNA

The YTHDF2 ‘reader’ protein has been demonstrated to increase RNA degradation via the direct binding of m^6^A-methylated transcripts and promotion of their transfer to p-bodies for degradation [31]. Our results suggests that *CD40L* mRNA degradation is mediated by m^6^A methylation. Next, we wanted to determine whether or not YTHDF2 is involved in the destabilization of methylated *CD40L* mRNA, so we analyzed m^6^A CLIP sequencing data available online [25] from mouse CD4^+^ T cell mRNA. As hypothesized based on the data presented in Figure 4, YTHDF2 can associate with *Cd40lg* mRNA at the 3′ UTR at the location that is also m^6^A-methylated. These data suggest that YTHDF2 can directly bind to m^6^A sequences on *CD40L* mRNA, thereby promoting the degradation of *CD40L* mRNA via YTHDF2-mediated mRNA destabilization.

## 4. Discussion

Despite the increasing understanding of the role that m^6^A RNA modifications play in various biological processes, the implications of m^6^A in CD4^+^ T cell regulation remain incompletely understood. In this study, we provide evidence for a novel role of m^6^A in CD4^+^ T lymphocyte activation and function. We observed the m^6^A enrichment of many transcripts involved in TCR activation upon CD4^+^ T cell activation. Specifically, m^6^A enrichment on *CD40L* mRNA is increased upon the activation of CD4^+^ T lymphocytes. CD40L is an important co-stimulatory protein expressed on the cell surface of CD4^+^ T cells. The CD40–CD40L interaction is directly involved in autoimmune disease pathogenesis [24], mainly in diseases that are driven by autoantibodies such as rheumatoid arthritis (RA) [32], systemic lupus erythematosus (SLE) [33] and Sjögren’s syndrome [33]. Several studies have implicated m^6^A and its regulatory proteins to be involved in autoimmune disease pathogenesis [34,35,36,37,38,39,40]. However, the expression of CD40L and a possibly compromised CD40–CD40L interaction were not evaluated in these studies. Increasing our understanding of the regulation of CD40L expression could contribute to a better understanding of the disease mechanisms underlying these autoimmune diseases. Here, we demonstrate that the manipulation of m^6^A in T cells, either by inhibiting the activity of FTO or METTL3 directly affects CD40L expression levels. This research also suggests that m^6^A ‘reader’ YTHDF2 is probably able to bind to m^6^A-specific sequences on CD40L mRNA to regulate its expression. Further research needs to be performed to find the exact mechanism behind this regulatory mechanism. In this study, meRIP sequencing was used to map m^6^A enrichment in CD4^+^ T lymphocytes. There are some drawbacks to this approach. First, our experiment was a single experimental replicate, and therefore only provides a rough overview of the methylated transcript that needs to be validated, as we did for CD40L. In addition, the resolution of meRIP sequencing is limited [41]. Therefore, it is not possible to exactly determine the localization of detected m^6^A modifications and to quantitatively measure the fraction of transcripts that contain this modification. Also, the m^6^A antibodies that are used for meRIP sequencing do not exclusively detect m^6^A, but can also detect the N6,2′-O-dimethyladenosine (m^6^A_m_) modification that is located at the 5′ UTR of mRNA transcripts [4,42]. Since m^6^A_m_ is present in a lower abundance compared to m^6^A, and considering the known location of m^6^A_m_ at the 5′ UTR, the presence of these modifications can still be distinguished. Due to the limited resolution with meRIP-seq, it is not possible to determine the exact adenosine(s) in the CD40L transcript that is/ are methylated. It would be valuable to exactly determine the location of the m^6^A modification(s) on *CD40L* mRNA in the future with higher-resolution sequencing approaches such as m^6^A-CLIP [4]. The additional experiments can be performed where the DRACH-sequences containing the methylated adenosine(s) can be mutated to further explore this process and determine the location of the adenosine that is most dominant for the observed phenotype.

Transcriptome-wide m^6^A mapping in previous studies revealed that roughly one of every three mRNA transcripts in human cells contain m^6^A modifications [1,3]. These transcripts contain three to five m^6^A-specific DRACH sequences on average, situated at various locations within the transcript. For the vast majority of transcripts, only a part of the DRACH sequences are methylated [1]. Using SRAMP [43], a computational predictor of mammalian m^6^A sites, six high-confidence m^6^A consensus sequences were identified in human *CD40L* mRNA. It would therefore be interesting to establish a direct correlation between a specific m^6^A modification on the *CD40L* transcript, mRNA faith and therefore CD4^+^ T cell activation. This could be achieved via point mutations of specific adenosines in the DRACH sequences of *CD40L* mRNA, thereby specifically reducing the amount of m^6^A modifications and assessing the effects on the abundance of expressed CD40L protein.

CD40L is not the only co-stimulatory molecule expressed on activated CD4^+^ T cells, and co-inhibitory molecules contribute to the regulation of an immune response. Therefore, it is plausible that the expression of more of these molecules would be regulated in similar ways. In our meRIP-seq data, many m^6^A enrichment changes were observed upon the activation of CD4^+^ T lymphocytes. Methylation changes could not only be observed for co-stimulatory molecule CD40L, but also for CD28 and inducible T cell co-stimulators (ICOS). Co-inhibitory proteins that showed m^6^A enrichment changes are programmed cell death-1 (PD-1), cytotoxic T- lymphocyte antigen-4 (CTLA-4) and CD45. Although we did not examine the methylation of these transcripts in detail, these observations indicate that the regulation of co-stimulatory/inhibitory protein expression via m^6^A is not an exclusive regulatory mechanism for CD40L.

We hypothesize that the m^6^A-mediated regulation of CD40L expression is a constant negative feedback mechanism in order to regulate the intensity and duration of an immune response. In the early immune response, CD40L expression is upregulated to enable co-stimulation and TCR activation. In later stages of inflammation, CD40L expression is reduced via m^6^A-mediated regulation to resolve the immune response. This proposed regulatory mechanism would prevent the chronic overactivation of CD4^+^ T lymphocytes, potentially leading to tissue damage and severe immunopathology.

## 5. Conclusions

Our data increase the general understanding of the role of m^6^A in CD4^+^ T lymphocyte activation and function. As CD40L is a co-stimulatory protein expressed by CD4^+^ T cells and the CD40–CD40L interaction is implicated in the pathogenesis of several diseases, this study contributes to our general understanding of autoimmune disease pathology.

## Figures and Tables

**Figure 1 biology-12-01004-f001:**
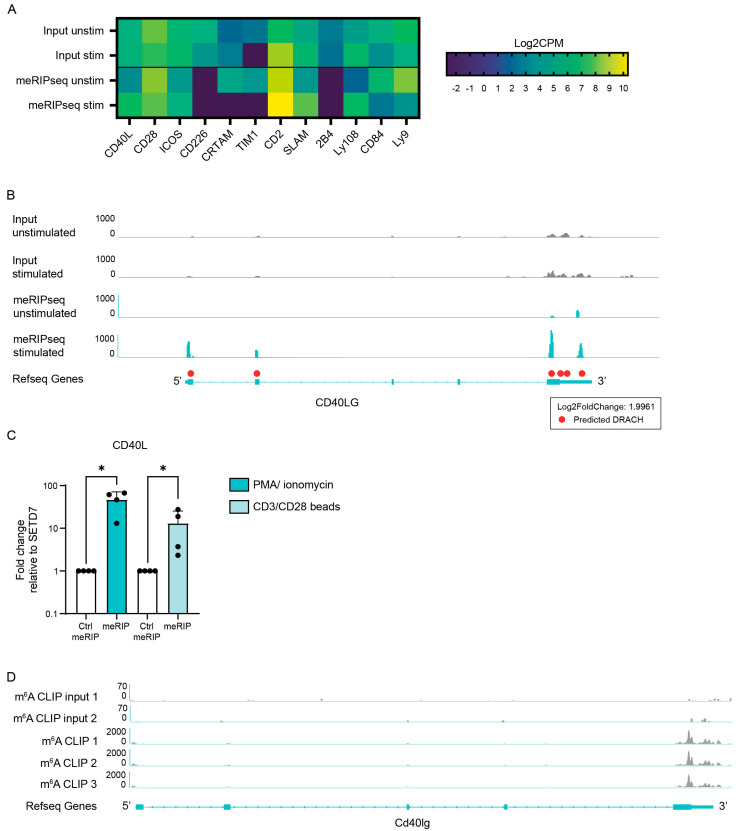
Increase in m^6^A enrichment on *CD40L* mRNA in activated CD4^+^ T lymphocytes. (**A**) Heat map of m^6^A enrichment changes in mRNA transcripts of co-stimulatory molecules, expressed in Log2CPM. meRIP sequencing data on unstimulated CD4^+^ T lymphocytes and CD4^+^ T lymphocytes stimulated with CD3/CD28 beads overnight (*n* = 1). (**B**) meRIP sequencing in healthy control CD4^+^ T lymphocytes. Cells were activated with CD3/CD28 beads (*n* = 1). (**C**) meRIP qPCR in healthy control CD4^+^ T lymphocytes. Cells were activated with either 4 h PMA and ionomycin or overnight with CD3/CD28 beads. *p* values were calculated using a Mann–Whitney *t*-test (* *p* < 0.05) (*n* = 4). (**D**) Integrative Genomics Viewer tracks of m^6^A CLIP read distribution in naïve mouse T lymphocytes provided by Ito-Kureha et al. [25]. Data obtained from three biological replicates.

**Figure 2 biology-12-01004-f002:**
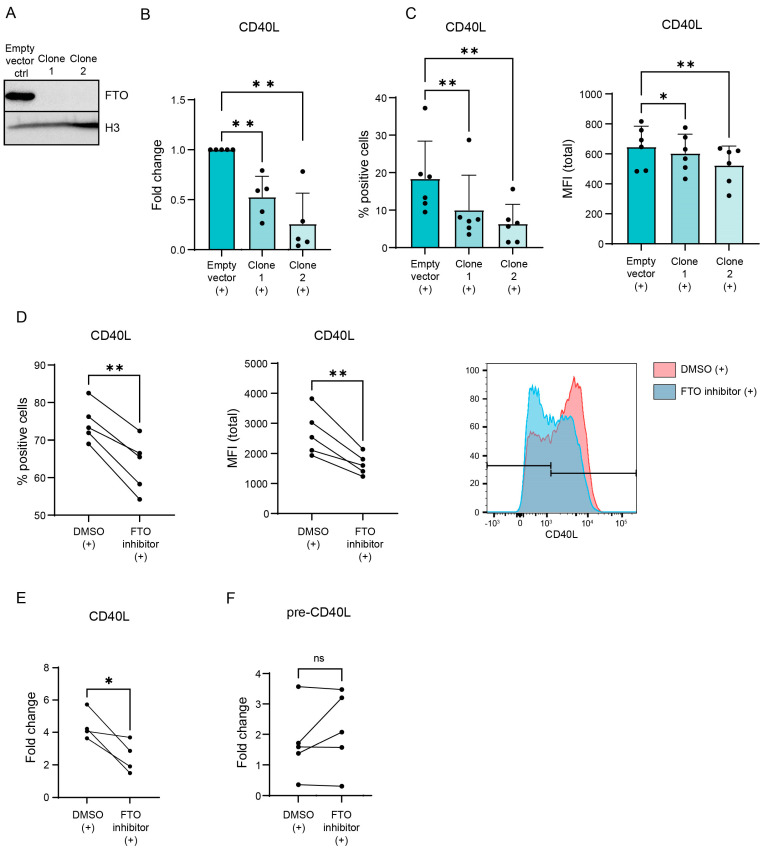
Regulation of CD40L expression via m^6^A ‘eraser’ FTO in activated CD4^+^ T lymphocytes. (**A**) CRISPR/Cas9 knockout of *FTO* in Jurkat cells compared to control. Western blot for FTO in two single-cell clones. (**B**) Jurkat cells with knockout of FTO activated with 4 h of PMA and ionomycin. qPCR on *CD40L* mRNA relative to housekeeping gene *GUSB*. (**C**) Jurkat cells with CRISPR/Cas9 knockout of *FTO* activated with 4 h PMA and ionomycin. Flow cytometry on CD40L protein expression. (**D**) Healthy control CD4^+^ T lymphocytes pre-incubated for 48 h with 50 µM entacapone. The cells were subsequently activated with CD3/CD28 beads overnight. Flow cytometry was performed on the CD40L protein. Right: representative histograms of CD40L protein reduction after entacapone treatment. (**E**) Healthy control CD4^+^ T lymphocytes pre-incubated for 48 h with 50 µM entacapone. The cells were subsequently activated with CD3/CD28 beads overnight. qPCR on *CD40L* mRNA. (**F**) Healthy control CD4^+^ T lymphocytes pre-incubated for 48 h with 50 µM entacapone. The cells were subsequently activated with CD3/CD28 beads overnight. qPCR on pre-*CD40L* mRNA. *p* values were calculated using a paired *t*-test; not significant (ns), * *p* < 0.05, and ** *p* < 0.01.

**Figure 3 biology-12-01004-f003:**
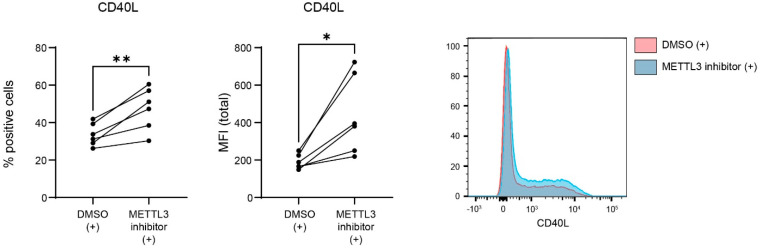
Regulation of CD40L expression by METTL3 in activated CD4^+^ T lymphocytes. Healthy control CD4^+^ T lymphocytes were incubated for 48 h with 30 µM STM2457, and subsequently activated with CD3/CD28 beads overnight. Flow cytometry on the CD40L protein and representative histograms of CD40L protein increase after STM2457 incubation. *p* values were calculated using a paired *t*-test; * *p* < 0.05, and ** *p* < 0.01.

**Figure 4 biology-12-01004-f004:**
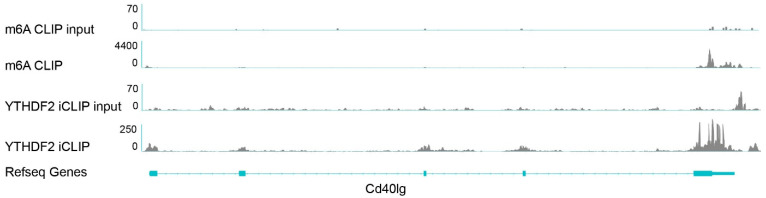
Possible binding of the m^6^A reader YTHDF2 to m^6^A sequences on *Cd40lg* mRNA. Integrative Genomics Viewer tracks of m^6^A CLIP and Ythdf2 iCLIP read distribution in naïve mouse T lymphocytes. Data combined from three biological replicates of the m^6^A CLIP and two replicates for the YTHDF2 iCLIP.

## Data Availability

The meRIP sequencing data have been deposited online. Read count data and per gene TMM normalized log2CPM expression values from meRIP RNA-seq are accessible at 10.5281/zenodo.8016876. Not all raw RNA sequencing data used in this study are publicly available due to research participant privacy/consent. Therefore, these data are only available upon request and after signing a Data Sharing Agreement within a specially designed UMC Utrecht-provided environment.

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
