# Peer review of "N6-Methyladenosine Directly Regulates CD40L Expression in CD4+ T Lymphocytes"

_biology, 2023, doi:10.3390/biology12071004_

Round 1
Reviewer 1 Report
The authors identify the role of the post-transcriptional N6-Methyladenosine modification on regulation of CD40L expression in CD4+T lymphocytes. They base their study on data obtained from a meRIP sequencing experiment, were CD4L was identified as an m6 enriched gene. These results are relevant as they might explain CD4+T cell activation mediated by CD40L m6 methylation.
Although interesting, there are some unclear issues with this study that should be clarified:
1 – Regarding the meRIP sequencing data, could you provide the accession numbers that were deposited in the databases? Also, would it be possible to have an excel file with all the differentially analysed genes so that everyone could acess this information that might be relevant for other studies.
2 - Why only CD4L was analysed? It is understandable that it is not possible to analyse all the genes simultaneously but can you explain briefly in what evidence you suggested to follow CD4L and not the other molecules? Also, can you clarify how the selection of genes present in figure 1A was done? Was it based on fold change differences? Are these the top differentially expressed genes relative to control samples?
3 - The qPCR results in figure 1C should be presented as relative fold change (relative to the control meRIP and reference gene – B2M, GUSB?) using the formula 2^-DDCT and not as CT values. The differences observed will be more trustable and more evident if you use a normalizing gene.
4 - Which guide RNA (sequence) was used for the CRISPR/CAS manipulation? Nothing is described about this method. The authors should include more information on these experiments.
5 – Can the authors indicate if there are some studies on M6A or FTO levels in patients with autoimmune diseases where CD40:CDF40L interactions are compromised?
Reviewer 2 Report
Vroonhoven et al aims to uncover m6A marks deposited upon activation of CD4+ T lymphocytes. To this end, they isolate PBMCs from the blood of healthy donors (n=3) and activate T cells with PMA/I or CD3/CD28. They then carry out meRIP-seq with mRNAs isolated from unstimulated and stimulated cells. By using FTO CRISPR knockout cells and METTL3 inhibitors, they interrogate the dependence of the stability of a candidate transcript, CD40L, on FTO knockout or METTL3 inhibitor. They carry on with the bioinformatics analyses of some CLIP-seq data to identify a potential reader protein. They report that 687 transcripts are differentially m6A methylated upon activation of PMBC cells. They select CL40L for further characterization. The authors generate an FTO KO jurkat cell line and show that CD40L expression and the percentage of CL40L+ cells decrease upon FTO knockdown. Congruently, the use of METTL3 inhibitor leads to an increase in the percentage of CD40L+ cells. From a mouse dataset, they identify bioinformatically YTHDF2 as a potential reader protein that interacts with CD40L transcript.
I believe that the authors present convincing evidence that CD40L transcript abundance is modulated by the m6A epitranscriptomic machinery, FTO and METTL3 in particular. The data is presented clearly. I would like to raise the following points for consideration to improve the quality of the manuscript prior to publication.
Major points:
1. Were the meRIP-seq data deposited to a public database? I believe that it must be!
2. Figure 1: Please include all three biological replicates in the heat maps and the IGV shots so we can see the variations. Figure 1B (lines 158-189): Please state the fold of enrichment quantitatively. Also on the figure, show the location of DRACH or any other m6A motifs that might exist.
3. Line 180: please cite a reference for this statement.
4. Figure 3: How did the author ascertain that the METTL3 inhibitor in fact inhibited METTL3 activity? Any positive control?
5. Figure 4: Lines 238-239: the bioinformatics analysis of the CLIP-seq data is overstated in lines 238-240. The CLIP-seq data only suggests that Ythdf2 binds to CD40L mRNA in mice. We cannot automatically assume that it recognizes an m6A mark. The authors do not provide any data for this. Additionally, the authors do not provide any data in human cells. I think that the statements that “YTHDF2 directly recognizes m6A-containing sequences or directly affects its expression..etc” are without any basis; and extremely overstated in abstract, discussion and throughout the results.
6. Discussion: The authors should discuss the drawback of meRIPseq approaches, e.g., the resolution of the information collected. Also, the number of potential m6A consensus sequences on CD40L should be discussed. It would also be nice to propose how to establish a direct correlation between a specific m6A mark and CD40L abundance and T cell activation as a future perspective, especially if the authors identify more than one potential m6A consensus sequence.
7. On several occasions, m6A RNA methylation is described as a post-transcriptional process (e.g., lines 14 and 50). However, there are numerous studies reporting co-transcriptional modifications. Please revise accordingly.
Minor points:
1. Lines 54-55: This statement needs to be revised. The current version implies a stop codon in the 5’ UTR.
2. Please convert all “M6A”s to “m6A”
3. Line 57, please write the full name of METTL14 in its first use. Also, CD40L is abbreviated several times throughout the manuscript. Please adopt the universal rules in abbreviating gene names.
4. In many cases, “hyphens” are misused. Please do not leave space right after the use of a hyphen. Please convert all “heat- activated” into “heat-activated”. It would be nice to check the whole manuscript for all such cases.
5. Line 110; “manufacturers” to “manufacturer’s”
6. Line 123, “reads that were not multi-mapped” to “unique reads”
7. Lines 143-145, this part could be integrated into other parts where appropriate.
Moderate editing is needed.
Round 2
Reviewer 1 Report
Dear authors,
Thank you for your reply and your clarifications.
I am ready to accept the paper provided this journal policy agrees with your reply:
"We agree with the reviewer that the information should be available for the scientific community. We have deposited all read count data at Zenodo for analyses by others. The raw reads from the sequencing experiment cannot be shared online due to research participant privacy/consent, but are available upon request. As requested by the reviewer we have now also deposited an excel file with the fold changes of all transcripts at Zenodo. We have adapted and extended the manuscript to include all the information on where to find these datasets".
Author Response
We thank the reviewer and are pleased to read that he/she is now ready to accept our manuscript for publication in Biology.
Reviewer 2 Report
Vroonhoven et al aims to uncover m6A marks deposited upon activation of CD4+ T lymphocytes. To this end, they isolate PBMCs from the blood of healthy donors (n=3) and activate T cells with PMA/I or CD3/CD28. They then carry out meRIP-seq with mRNAs isolated from unstimulated and stimulated cells. By using FTO CRISPR knockout cells and METTL3 inhibitors, they interrogate the dependence of the stability of a candidate transcript, CD40L, on FTO knockout or METTL3 inhibitor. They carry on with the bioinformatics analyses of some CLIP-seq data to identify a potential reader protein. They report that 687 transcripts are differentially m6A methylated upon activation of PMBC cells. They select CL40L for further characterization. The authors generate an FTO KO jurkat cell line and show that CD40L expression and the percentage of CL40L+ cells decrease upon FTO knockdown. Congruently, the use of METTL3 inhibitor leads to an increase in the percentage of CD40L+ cells. From a mouse dataset, they identify bioinformatically YTHDF2 as a potential reader protein that interacts with CD40L transcript.
It appears that the authors have addressed all the points raised to their best ability. Although the revision has improved the manuscript, I am affraid that the major concerns have not been adressed properly. Specifically,
1. The authors must make data accessible to the readers, citing its accession number in the Manuscript.
2. More importantly, a single replicate of a MeRIP is highly prone to introduce many errors without proper statistical analyses. I would at least expect to state in the manuscript that the data come from a single replicate and provide convincing evidence about the suitability of the data in its current form.
3. I'd have liked to see a western blot analysis of METTL3 after the use of its inhibitor, rather than citing a reference.
Minor typing errors to be fixed.
Author Response
We would like to that the reviewer for the feedback and are happy to clarify:
- The authors must make data accessible to the readers, citing its accession number in the Manuscript.
We agree. All sequencing read counts are deposited at Zenodo under accession nr. 8016876 allowing for analyses by other researchers. Due to the very strict privacy legislation of our institute we are unable to share the raw sequencing reads online, however these are all available upon request. This information about data availability including the accession number can be found at the bottom of the manuscript under the heading “Data Availability Statement”.
- More importantly, a single replicate of a MeRIP is highly prone to introduce many errors without proper statistical analyses. I would at least expect to state in the manuscript that the data come from a single replicate and provide convincing evidence about the suitability of the data in its current form.
We agree with the reviewer and that is also the reason that we focus on one specific transcript (CD40L) and do not make strong claims about the entire transcriptome. Since we validated CD40L methylation by QPCR, online datasets, and functional experiments we are confident in the CD40L data. The manuscript already contained information that this specific experiment is a single replicate (in figure legend), but we have now further elaborated on this in the discussion section.
- I'd have liked to see a western blot analysis of METTL3 after the use of its inhibitor, rather than citing a reference.
We understand the question and it was initially our intention to perform exactly this experiment. However, the inhibitor does not affect the protein expression of METTL3, but it binds the METTL3/METTL14 protein complex and thereby inhibits its activity. Therefore the Western blot will not add any information. The manuscript includes three references of studies that use this inhibitor for inhibition of METTL3 function.
- Minor typing errors to be fixed.
We apologize for this. We have checked the manuscript again and edited some typing errors.
Round 3
Reviewer 2 Report
The authors have addressed the concerns properly. The revised discussion and the deposition of the data have eliminated all the concerns. I believe that the manuscript warrants publication at its current form.